# Physiological Offset Parameters of the Adult Shoulder Joint—A MRI Study of 800 Patients

**DOI:** 10.3390/diagnostics12102507

**Published:** 2022-10-16

**Authors:** Marc-Pascal Meier, Lars Erik Brandt, Dominik Saul, Paul Jonathan Roch, Friederike Sophie Klockner, Ali Seif Amir Hosseini, Wolfgang Lehmann, Thelonius Hawellek

**Affiliations:** 1Department of Trauma Surgery, Orthopaedics and Plastic Surgery, University Medical Center Goettingen, Robert-Koch-Straße 40, 37075 Göttingen, Germany; 2BG Trauma Center Tuebingen, Department of Trauma and Reconstructive Surgery, Eberhard Karls University Tuebingen, 72076 Tuebingen, Germany; 3Kogod Center on Aging and Division of Endocrinology, Mayo Clinic, Rochester, MN 55905, USA; 4Department of Diagnostic and Interventional Radiology, University Medical Center Goettingen, Robert-Koch-Straße 40, 37075 Göttingen, Germany

**Keywords:** humeral offset, glenoidal offset, MRI, shoulder morphology, shoulder arthroplasty

## Abstract

**Highlights:**

800 shoulder MRIs were retrospectively evaluated for humeral, glenoidal and latero-glenoidal humeral offset, as well as cortical and humeral shaft axis offset.Significantly higher values for all analysed offset parameters were found in male shoulder joints compared to female shoulder joints.Significant side-specific differences were found for humeral, glenoidal and latero-glenoidal humeral offset.A correlation of the offset parameters with age or the osteoarthritis grade exist only for cortical offset.These results should be considered in shoulder diagnostics and surgery.

**Abstract:**

***Background*:** Humeral offset (HO) and glenoidal offset (GO) are important morphological parameters in diagnostics and therapy for shoulder pathologies. However, physiological reference values have not yet been sufficiently determined. The aim of the present study was to establish physiological reference values for shoulder offset parameters (SOPs). ***Methods*:** MRI images of the shoulder joints of 800 patients (mean age: 50.13 years [±16.01]) were analysed retrospectively. HO, GO, lateral glenoidal humeral offset (LGHO), humeral shaft axis offset (HAO) and cortical offset (CO) were measured. SOPs were examined for associations with age, gender, side and osteoarthritis. ***Results*:** The mean HO was 26.19 (±2.70), the mean GO was 61.79 (±5.67), the mean LGHO was 54.49 (±4.69), the mean HAO was 28.17 (±2.82) and the mean CO was 16.70 (±3.08). For all SOPs, significantly higher values were measured in male shoulders. There was a significantly (*p* < 0.001) higher mean value for HO, GO and LGHO in right shoulders. There was a significant correlation between age and LGHO, and HAO and CO, but not between age and HO or GO. Shoulders with osteoarthritis and non-osteoarthritis did not differ in the mean value of HO, GO, LGHO and HAO, except for CO (*p* = 0.049). ***Conclusion*:** Reference values for SOPs in the adult shoulder joint were determined for the first time. Significant gender-specific differences were found for all measured SOPs. In addition, it was seen that for some SOPs, the joint side and the patient’s age has to be taken into account in shoulder diagnostics and surgery.

## 1. Introduction

Humeral offset (HO) and glenoidal offset (GO) parameters are important morphological parameters in diagnostics and in therapy for shoulder pathologies. Humeral head preserving surgery remains the current state-of-the-art surgery for younger patients in the case of a fracture [1,2]. Likewise, arthroplasty of the shoulder joint is increasingly gaining importance in the treatment of degenerative and traumatic shoulder pathologies, especially in older patients [3,4,5,6]. In order to achieve an optimal postoperative result, the best possible reconstruction of the physiological shoulder morphology is required for both joint-preserving and joint-replacing therapy [7,8]. However, this requires valid ranges of physiological reference values for preoperative diagnostics and planning. There is a consensus that cross-sectional imaging is required for the optimal radiological assessment of shoulder morphology [9,10,11]. In its function as a spherical joint, the shoulder joint has a very large range of motion [12,13]. This is precisely why an exact recording of the morphological interactions of the humeral head and the glenoid using a two-dimensional radiograph image is not sufficient [9,10,11]. In this context, reconstruction of the humeral offset (HO) and glenoidal offset (GO) has a major influence on undisturbed function after joint replacement [7,8,14]. A higher lateral glenoidal humeral offset (LGHO) in native shoulder joints is associated with better clinical function and a higher range of motion [4].

Depending on osteosynthesis or the arthroplasty techniques used, changes in shoulder offset parameters (SOPs) affect postoperative function in different ways [14]. If, for example, the actual physiological offset is increased by a total shoulder arthroplasty, this can lead to rotator cuff insufficiency and damage to the glenoid bone structure in the long term [7,15,16]. Consequently, LGHO reconstruction reduces the risk of early aseptic loosening in hemi- and total shoulder arthroplasty [7,16].

In contrast, increasing the GO in isolation in reverse shoulder arthroplasty results in better joint rotation and reduces the risk of scapula notching and bony impingement [7,16,17,18]. However, this leads to a higher risk of fractures of the acromion and a decrease in the efficiency of the deltoid muscle. Increasing the HO in isolation improves the efficiency of the deltoid muscle, but is associated with reduced joint stability. The risk of acromion fractures remains [6,16,17,18]. The risk of a secondary fracture of the shoulder joint due to an offset change must be considered critical. To the authors’ knowledge, the effects of possible offset changes in the case of humeral head therapy using osteosynthesis in the event of a fracture have not been reported so far.

Furthermore, there is no established measurement method in the current literature for the detection of GO in magnetic resonance imaging (MRI). Computed tomography (CT)-based methods are not directly transferable [14].

From the aforementioned, we conclude that SOPs are important factors for shoulder joint function. In particular, surgical therapy, such as humeral head osteosynthesis or shoulder arthroplasty, can change the SOPs, which can result in functional restrictions. In the current literature, there is no study that analyzes physiological humeral and glenoidal offset in a large, primarily radiologically healthy shoulder using cross-sectional imaging. The present study aims to answer the question of whether there are side-, age- and gender-specific differences in SOPs and whether the changes in SOPs are related to osteoarthritis grades. In addition, we aim to establish a new measurement method for recording GO using MRI. The aim of the present study was to create physiological reference values for the shoulder offset parameters in order to optimize the diagnostics and surgical therapies for shoulder pathologies. 

## 2. Materials and Methods

### 2.1. Patients

Between 2013 and 2021, a total of n = 2629 patients underwent magnetic resonance imaging of the shoulder joint in the Department for Diagnostic and Interventional Radiology at the University Medical Center Goettingen. These MRIs were reviewed retrospectively. After applying inclusion and exclusion criteria, 806 patients were included in the final analysis. The study participants were divided into two age groups (20–50 and >50 years). The study was approved by the local ethics committee (IRB number 17/5/22) and performed in accordance with the principles expressed in the Declaration of Helsinki. Without exception, the evaluated MRIs were taken as a part of routine diagnostics because the patients presented with clinical symptoms. In a standardized manner, all the patients were placed in a shoulder positioning tray on the MRI table. All the MRIs were assessed by a senior radiologist and LEB, MPM, ASAH and TH in order to exclude extended structural injuries or heavy joint degeneration.

### 2.2. Inclusion Criteria

All the examinations were accessible via PACS (Picture Archiving and Communication System) between 1 January 2013 to 31 December 2021 and were initially included in the study. Out of these, all patients aged 20 years or more were included. All the MRI scans were performed on patients to assess their shoulder joint pathologies. Only the patients with a Kellgren/Lawrence score [19] <3 were classified as “shoulder joint healthy” (SJH). The patients with a Kellgren/Lawrence Score of ≥3 were classified as “shoulder joint diseased” (SJD). All the MRIs were examined by the internal radiology department as part of the clinical diagnostic procedure. Every report was re-evaluated by LEB, MPM, ASAH and TH in a blinded fashion. 

### 2.3. Exclusion Criteria

All the patients with fractures, osteonecrosis, dysplasia or tumors were excluded. Similarly, the patients who had undergone osteosynthesis or arthroplasty were excluded. The data did not include patients who had any other implants after shoulder joint preservation surgery. The patients with held humeral head luxation and extensive injuries of the rotator cuff were also excluded. Low-quality MRIs (based on only a few gates), were excluded. In addition, all MRIs with imaging artefacts were ruled out.

### 2.4. MRI Analysis, Parameters and Methods of Measurement

All measurements were taken via PACS (Picture Archiving and Communication System). Software from GE Healthcare called Centricity^TM^ Universal Viewer was used (RA1000, edition 2019, Buckinghamshire, Great Britain). The osteoarthritis score for each shoulder joint was classified according to the Kellgren/Lawrence (KL, [17]) scoring system in order to group patients into SJH and SJD groups. The SOPs were measured using established methods. The HO, GO, LGHO, the humeral shaft axis offset (HAO) and the cortical offset (CO) were determined [14,20,21,22]. The GO was recorded according to a new measurement method that was developed by the authors and modified from an established method [14]. To measure the GO, the center of rotation of the humeral head was determined in the coronal view. Using measurement software, the position of the center of rotation was used to determine the corresponding point in the axial plane. The distance between this point and the end of the scapula neck defined the GO. Figure 1 shows the principles of this measurement methodology. All the radiographic parameters in this MRI study were manually and separately measured in a standardized manner by the same observer (LEB) under the supervision of an experienced senior radiologist (ASAH). The intraobserver reliability of the measurements for all the parameters was assessed for a subset of 50 subjects by blinded re-evaluation 2 weeks after the first measurement was taken, using the same technique. Interobserver reliability was assessed by two observers (LEB and MPM) independently for 50 subjects. 

### 2.5. Statistics

For the side-, age- and gender-specific analyses of HO, GO, LGHO, HAO and CO, a Mann–Whitney U test was used. The Mann–Whitney U test was implemented in order to compare the SJH and SJD groups for all the assessed OPs. In order to detect the possible interrelationships between the different OPs and the patients’ age, a Spearman correlation was set. Intraobserver and interobserver reliabilities were evaluated using intraclass correlation coefficients (ICC). Overall, the means ± standard deviations are stated. The statistical analyses were performed using GraphPad Prism 9.00 (GraphPad Software, San Diego, CA, USA), SPSS Statistics software version 27.0 (IBM SPSS Inc., Chicago, IL, USA) and Microsoft Excel (Microsoft Office 2016, Redmond, WA, USA). Significant differences are marked with asterisks (*** *p* < 0.001, ** *p* < 0.01, * *p* < 0.05).

## 3. Results

### 3.1. Characteristics of the Study Population

In this study, 449 males (56.13%) and 351 females (43.87%) were analyzed. The final analysis included 429 right shoulders (53.63%) and 371 left shoulders (46.37%). The mean age of the study population was 50.13 years (±16.01 [age range: 20–89 years]). 

### 3.2. Analysis of the Shoulder Offset Parameters

#### 3.2.1. Humeral Offset

In total (n = 800), a mean HO of 26.19 (±2.70 [18.8–36.4]) was found. The mean HO in left shoulders (n = 371) was 25.45 (±2.62) and the mean HO in right shoulders was 26.83 (±2.59). For the younger patients (20–50 years, n = 384), a mean HO of 26.07 (±2.61) was detected, whereas the older patients (>50 years, n = 416) had a mean HO of 26.31 (±2.77). The analysis yielded a mean HO of 24.17 (±2.01) in female shoulders (n = 351) and 27.77 (±2.03) in male shoulders.

#### 3.2.2. Glenoidal Offset

In total, a mean GO of 61.79 (±5.67 [42–81.1]) was found. The mean GO was 60.67 (±5.29) in left shoulders and 62.75 (±5.81) in right shoulders. For the younger age group, a mean GO of 62.13 (±5.41) was detected, whereas the older patients showed a mean GO of 61.47 (±5.89). The analysis yielded a mean GO of 58.14 (±4.52) in female shoulders and 64.64 (±4.78) in male shoulders.

#### 3.2.3. Lateral Glenoidal Humeral Offset

In total, a mean LGHO of 54.49 (±4.69 [42.2–66.8]) was found. The mean LGHO was 53.74 (±4.54) in left shoulders and 55.13 (±4.72) in right shoulders. For the younger age group, a mean LGHO of 54.16 (±4.62) was detected, whereas the older patients showed a mean LGHO of 54.79 (±4.73). The analysis yielded a mean LGHO of 50.56 (±3.14) in female shoulders and 57.56 (±3.15) in male shoulders.

#### 3.2.4. Humeral Shaft Axis Offset

In total, a mean HAO of 28.17 (±2.82 [12.7–36.3]) was found. The mean HAO on the left shoulder joint was 28.14 (±2.93) and the mean HAO on the right joint side was 28.20 (±2.73). For the younger age group, a mean HAO of 28.17 (±2.80) was detected. The older patients showed a mean HAO of 28.17 (±2.85). The analysis yielded a mean HAO of 26.34 (±2.14) in female shoulders and 29.60 (±2.44) in male shoulders.

#### 3.2.5. Cortical Offset

In total, a mean CO of 16.70 (±3.08 [2.8–27.8]) was found. The mean CO in left shoulders was 16.74 (±3.21) and the mean CO in right shoulders was 16.67 (±2.97). For the younger age group, a mean CO of 16.99 (±2.96) was detected, whereas the older patients showed a mean CO of 16.44 (±3.17). The analysis yielded a mean CO of 15.92 (±2.80) in female shoulders and 17.31 (±3.16) in male shoulders.

### 3.3. Analysis of Side Specific Differences for SOPs

There was a significant difference (*p* < 0.001) in the HO and GO between left (HO: 25.45 [±2.62]; GO: 60.67 [±5.29]) and right (HO: 26.83 [±2.59]; GO: 62.75 [±5.81]) shoulder joints. Similarly, a significantly (*p* < 0.001) higher mean LGHO was found in right shoulder joints (right: 55.13 [±4.72]; left: 53.74 [±4.54]). No side-specific differences were found for HAO (*p* = 0.609) and CO (*p* = 0.843). All the results are summarized in Table 1.

### 3.4. Analysis of Age-Specific Differences and Age Dependent Correlation Analysis for SOPs

A significant difference was found only for the CO between the age groups (*p* = 0.011). The analysis showed a mean CO of 16.99 (±2.96) in patients with an age of 20–50 years and a mean CO of 16.44 (±3.17) in patients aged over 50 years. There were no significant differences for HO (*p* = 0.363), GO (*p* = 0.124), LGHO (*p* = 0.081) and HAO (*p* = 0.756). These results are summarized in Table 2. 

The correlation analysis showed weak significant inverse correlations between age and LGHO (r_S_ = −0.089, *p* = 0.005), and CO (r_S_ = −0.082, *p* = 0.021). There were no significant correlations between age and HO (*p* = 0.134), GO (*p* = 0.061) or HAO (*p* = 0.637). The results of the correlation analysis are summarized in Figure 2. 

### 3.5. Analysis of Gender-Specific Differences for SOPs

For all SOPs, significantly (*p* < 0.001) larger measured values were found in male shoulder joints. The mean HO in female patients was 24.17 (±2.01) and the mean HO in male patients was 27.77 (±2.03). A larger GO was found in male shoulder joints (64.64 [±4.78]) compared to female shoulder joints (58.14 [±4.52]). In addition, there were higher values for mean LGHO and HAO in male shoulder joints (LGHO: 57.56 [±3.15]; HAO: 29.60 [±2.44]) compared to female shoulder joints (LGHO: 50.56 [±3.14], HAO: 26.34 [±2.14]). There was a mean CO of 17.31 (±3.16) in male shoulder joints and a mean CO of 15.92 (±2.80) in female shoulder joints. These results are summarized in Table 3. 

### 3.6. Analysis between SOPs and Grade of Osteoarthritis

In this study, 777 (97.13%) patients were graded as KL 0-2 (SJH) and 23 (2.87%) patients were graded as KL 3-4 (SJD). There was a significant difference (*p* = 0.049) between the mean CO of the SJH group (16.75 [±3.04]) compared to the SJD group (15.09 [±3.92]). The other SOPs showed no significant differences between these two groups (HO: *p* = 0.941; GO: *p* = 0.148; LGHO: *p* = 0.386; HAO: *p* = 0.571). The results are summarized in Table 4.

### 3.7. Analysis of Intraobserver and Interobserver Reliability

The ICCs for intraobserver reliability ranged from 0.94 to 0.99 and the ICCs for interobserver reliability ranged from 0.93 to 0.99, indicating excellent reliability. Taking into account the initial measurements, the control measurements by the same examiner and the measurements by a second examiner, cumulative ICC values of 0.94 to 0.99 (HO: 0.96; GO: 0.94; LGHO: 0.99; HAO: 0.98; CO: 0.97) resulted. These results are summarized in Figure 3.

## 4. Discussion

Shoulder offset parameters are important morphological parameters for diagnostics and therapies for shoulder pathologies [7,8,14,23]. Therefore, it is necessary to establish radiological reference values for the shoulder offset parameters in order to make correct diagnostic and therapeutic decisions for shoulder pathologies based on these data [9,10,11]. In the present study, the MRI scans of 800 shoulder joints were examined for the first time in order to establish reference values for future diagnostic and therapeutic applicability. 

The analysis of side-specific differences showed significantly higher HO, GO and LGHO measurements on the right joint side compared to the left joint side. To the authors’ knowledge, there is currently no other study that describes the side-dependence of HO, GO and LGHO in a primarily radiologically low osteoarthritis population (97.1%) of native shoulder joints. 

A possible reason is that 90% of the population is right-handed, and therefore uses the right upper limb more dominantly [24,25]. However, due to the retrospective nature of the present study, this assumption can only be made hypothetically. In order to finally clarify this question, additional studies that specifically examine the HO, GO and LGHO between left- and right-handers are needed. The clinical consequence of these results are that, in contrast to the hip joint [26,27], for the surgical treatment of severe degenerative diseases or fractures in the shoulder joint, the healthy contralateral side cannot be used uncritically for planning the offset reconstruction. This hypothesis must be supported by further studies. However, data from the present study can be used as a physiological reference to provide an important orientation for the reconstruction of the shoulder offset parameters. 

Age-dependent differences for the SOPs were not found, except for the CO. In addition, a weak inverse correlation between CO and patient age was found. Taking into account the osteoarthritis-dependent analysis, in which significantly lower measured values for CO were found in the group with a higher grade of osteoarthritis (KL 3-4), the most plausible reason for this is the loss of sphericity of the humeral head. Knowles et al. investigated the morphology of the humeral head between healthy shoulder joints and shoulder joints with osteoarthritis using 150 computer tomographic examinations [28]. They were able to demonstrate a loss of sphericity due to altered humeral head diameters. The resulting position of the humeral head, in relation to the glenoid, leads to changes in the medial cortical axis, which has an influence on the measured CO values. However, in this study, a sample that essentially did not show any high-grade osteoarthritis was deliberately examined in order to determine physiological reference values. In the sample examined, only 2,9% of patients (23/800) were classified with a Kellgren–Lawrence score [19] of >3, so the differences between the osteoarthritis group (23/800) and the non-osteoarthritis group (777/800) were regarded as critical due to the group size. Here, a future comparative study of the offset parameters collected in this study and the same measurements for a sample of shoulder joints with osteoarthritis would be useful. 

A weak correlation with patient age was also found for the LGHO. Significant differences between the age groups were not detected, so the correlations are classified as marginal. 

The analysis revealed significantly higher measurements for all SOPs in the male patients compared to the female patients. The gender-dependency of some parameters of shoulder morphology has already been reported in other studies. For example, Piponov et al. demonstrated a greater glenoid height and diameter in the coronal view for male patients, by retrospectively analyzing 108 shoulder CT scans [29]. Similarly, higher measurement values for glenoid height and width were reported by Mathews et al., who examined 18 body donors to detect potential sex-specific differences [30]. To the authors’ knowledge, this study is the first to describe gender-specific differences for the SOPs of the adult shoulder joint. In order to achieve reproducible measurement results, exact image morphological landmarks are necessary [14,31,32]. In everyday clinical practice, the quality of radiological imaging often differs, so that landmarks are partly difficult to identify. Several studies have already addressed the question of how morphological parameters of the shoulder joint can be reproducibly recorded [14,31,32,33]. Bodrogi et al. describes different measurement methods to record the humeral and glenoidal offset, by investigating 37 patients who underwent total shoulder arthroplasty, pre- and postoperatively [14]. The authors reached an inter-rater reliability of 0.8 for the glenoidal offset and an intra-rater reliability of more than 0.94. However, the methods used to measure glenoidal offset could not be applied to the MRIs examined in the present study because the coracoid and glenoid were rarely adequately represented in one image plane. Therefore, a new method of measuring GO was established in the present study. With regards to the intra- (0.94) and inter-rater (0.94) reliability (the cumulative ICC was 0.94), the method presented here for measuring GO is easily reproducible. This method uses the end of the shoulder neck as a reference point for determining the GO and, therefore, is based on a clear anatomical structure of the scapula, which can be reproduced in most of axial images of the shoulder joint in the MRI. Therefore, this should be considered for future studies investigating GO. In conclusion, the newly proposed method for the detection of GO in MRI scans is reproducible and, therefore, may be used in clinical practice in the future.

Humeral and glenoid offset parameters are important morphological parameters in diagnostics and therapies for shoulder pathologies [7,8]. The results of the present study represent a valid reference for the physiological offset parameters of the adult shoulder joint and should be taken into account in daily clinical practice. Especially in cases of severe fractures or terminal osteoarthritis, these physiological reference values are highly important in clinical practice as they represent an essential orientation with regards to offset reconstruction of the affected joint.

## 5. Limitations

A limitation of this study is that no comparisons were made between the MRI scans with related X-ray imaging. This was because many of the patients included in the study were only examined using MRI scans and not X-rays. A comparison between the different modalities seems useful. Therefore, the present study could have made even better comparisons with the results of other authors. In addition, comparisons between the measurements from MRI images and CT scans would be desirable. No biometric data were collected in the study and the retrospective nature of the study does not allow any conclusions to be drawn about the pain intensity of the patients. Further studies with a prospective design are necessary to verify the results of this study. The handedness of the patients was also not surveyed. This could also have an influence on the offset parameters. Studies dedicated to this question would also be of high scientific value. It should be noted that, based on the retrospective study design, no intra-individual bilateral measurements of the shoulder joints were performed. This means that the differences found in the mean values of the SOPs between the left and right shoulder could only be determined inter-individually and, therefore, have to be viewed critically. In order to make a statement about the possible intra-individual side differences for the SOPs, future prospective studies with bilateral joint side analyses should be carried out in order to confirm or revise this observation.

## 6. Conclusions

In the present study, the MRIs of 800 patients were examined for the first time in order to analyze the physiological reference values for SOPs in a sample of radiologically healthy shoulder joints. The mean values for HO, GO, LGHO, HAO and CO depending on gender, joint side and age were presented. HO, GO and LGHO showed significantly higher mean values on the right joint side. These results indicate that, in clinical practice, the healthy contralateral side cannot be used as a reference for surgical planning. However, further prospective studies are needed to support this thesis. For all offset parameters, significantly higher values were found in males compared to females. The age- and osteoarthritis-dependent analyses rendered significantly lower results for CO in older patients and patients with a higher grade of osteoarthritis, which seems to be caused by the loss of sphericity of the humeral head. These results should be considered in shoulder diagnostics and surgery. 

## Figures and Tables

**Figure 1 diagnostics-12-02507-f001:**
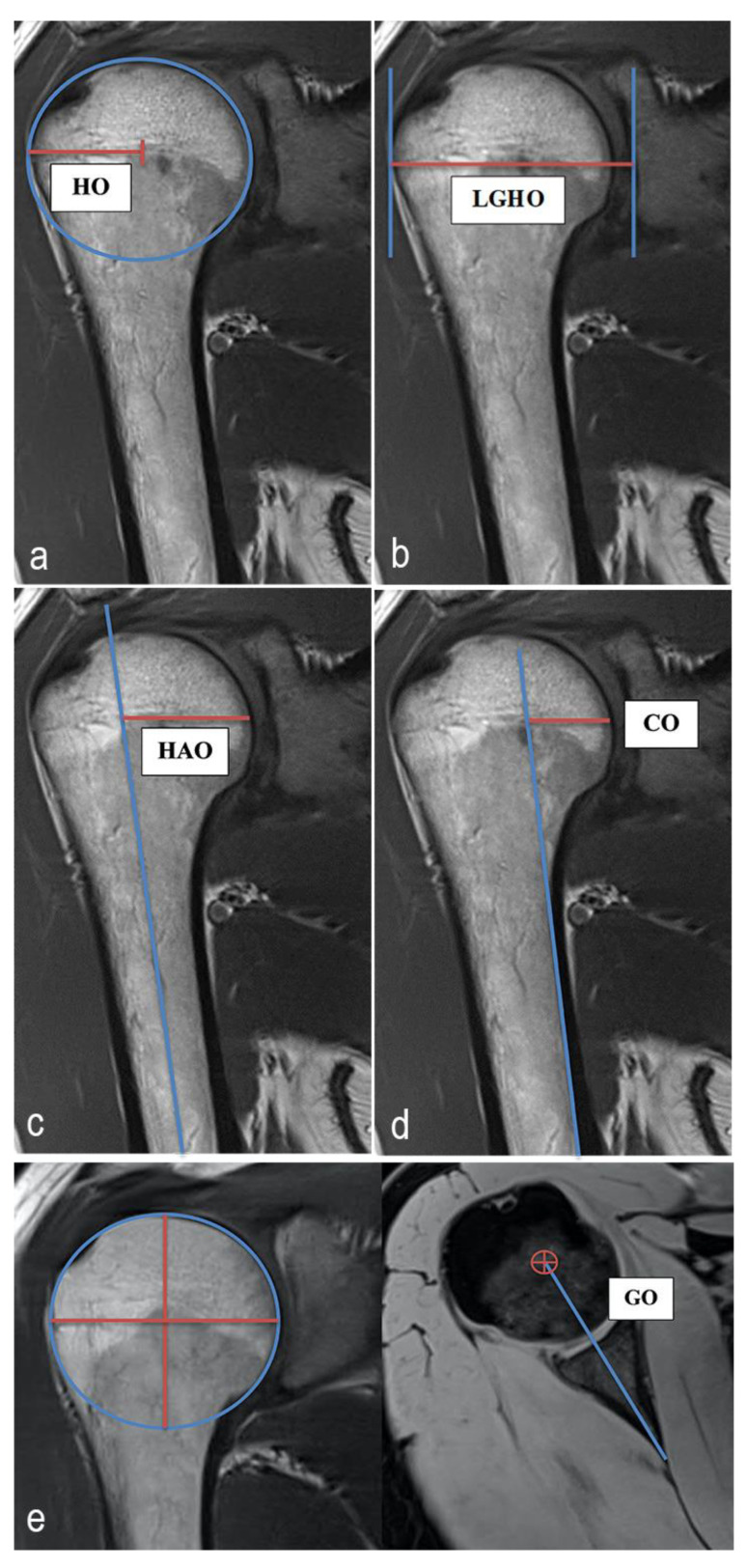
Exemplary depiction of the measurements of the humeral offset (HO, **a**), lateral glenoidal humeral offset (LGHO, **b**), humeral shaft axis offset (HAO, **c**), cortical offset (CO, **d**) and glenoidal offset (GO, **e**). The measurements of HO, LGHO, HAO and CO were all performed in the coronal view of the shoulder joint in the MRIs. GO was measured in the transversal view. To measure the HO, the centrum of rotation of the humeral head was determined. The distance from this point to the great tubercle defined the HO (**a**). The distance from the great tubercle to centr of the glenoidal joint surface described the LGHO (**b**). To determine the HAO, the distance between the humeral head shaft axis and the medial cortical humeral head was measured. The drawn horizontal line intersects the centrum of rotation of the humeral head (**c**). To measure the CO, the medial cortical bone of the humeral shaft was imaged. The distance from this axis to the medial cortical humeral head was defined as CO. The drawn horizontal line intersects the centrum of rotation of the humeral head (**d**). To measure the GO, the center of rotation of the humeral head was determined in the coronal view. Using measurement software, the position of the center of rotation was used to determine the corresponding point in the axial plane. The distance between this point and the end of the scapula neck defined the GO (**e**).

**Figure 2 diagnostics-12-02507-f002:**
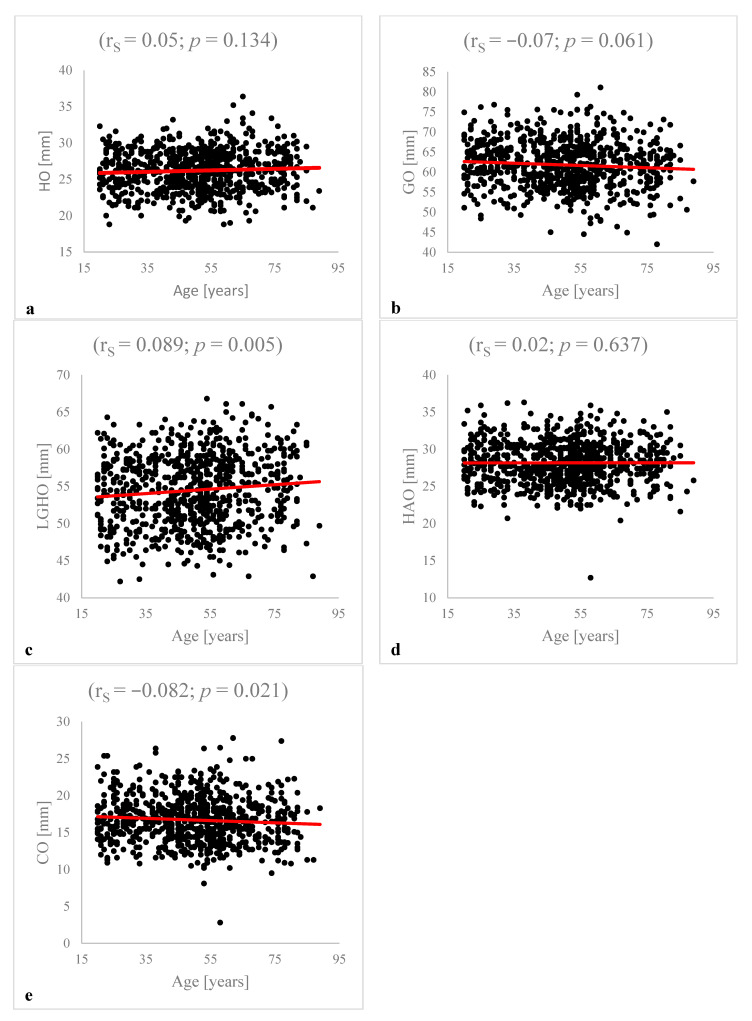
**Correlation analysis between patients’ age and HO, GO, LGHO, HAO and CO:** The analysis is based on Spearman correlations. There was no significant correlation between patient age and HO (r_S_ = 0.05; *p* = 0.134) (**a**), GO (r_S_ = −0.07; *p* = 0.061) (**b**) or HAO (r_S_ = 0.02; *p* = 0.637) (**d**). The analysis detected a significant correlation between LGHO (r_S_ = 0.089: *p* = 0.005) (**c**) and CO (r_S_ = −0.082; *p* = 0.021) (**e**) and patient age.

**Figure 3 diagnostics-12-02507-f003:**
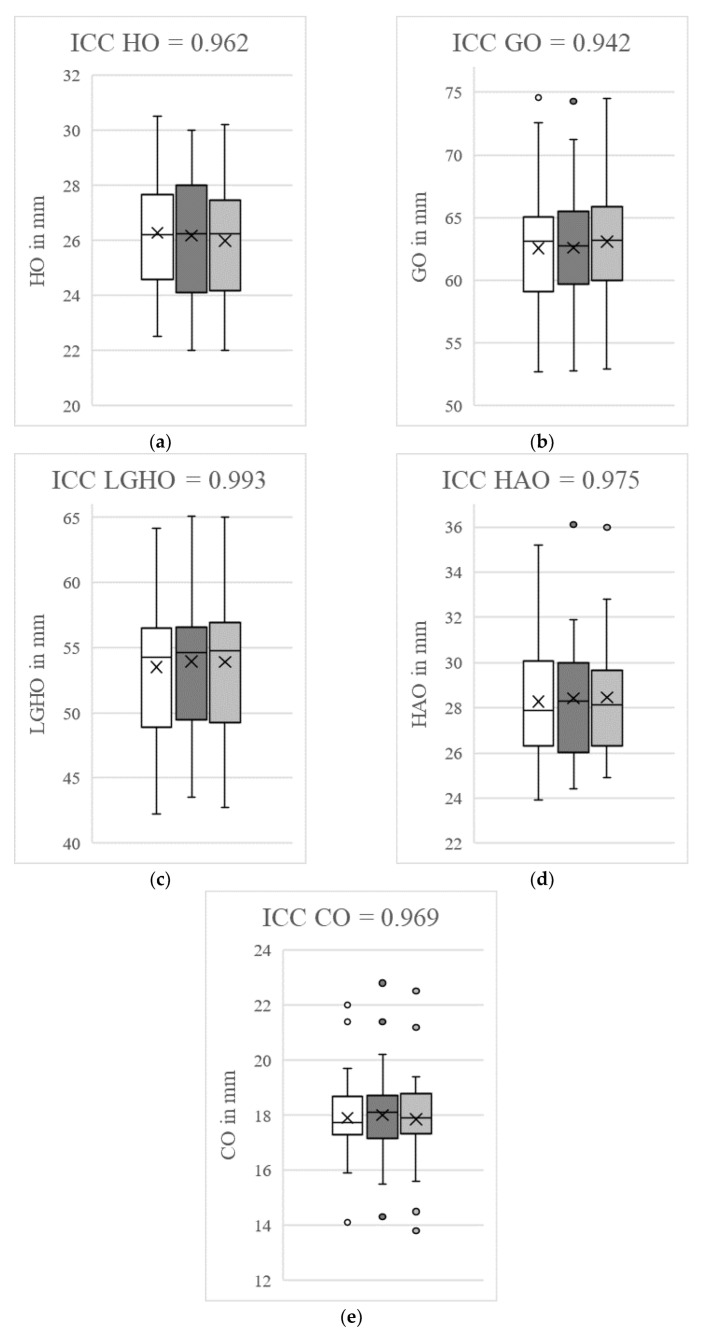
Determination of intraclass/interclass correlation coefficients (ICCs) for humeral offset (HO, **a**), glenoidal offset (GO, **b**), lateral glenoidal humeral offset (LGHO, **c**), humeral shaft axis offset (HAO, **d**) and cortical offset (CO, **e**). The white box plots represent the initial measurements by LEB and the dark grey box plots represent the second recordings by LEB. The box plots marked in light grey reflect the measurements by MPM. The figures show the cumulative ICCs between the initial and control measurements by LEB and the observations by MPM. The outliers are marked with points. Excellent measurement reliability (ICC > 0.9) was determined for all the offset parameters. The control measurements were carried out at intervals of two weeks by the same investigator and another investigator in a blinded fashion for 50 subjects.

**Table 1 diagnostics-12-02507-t001:** Analysis of side-specific differences between offset parameters.

		Total (n = 800)	Left (n = 371)	Right (n = 429)	*p*-Value
**HO**	[mm]	26.19 (±2.70)	25.45 (±2.62)	26.83 (±2.59)	<0.001 ***^,1^
**GO**	[mm]	61.79 (±5.67)	60.67 (±5.29)	62.75 (±5.81)	<0.001 ***^1^
**LGHO**	[mm]	54.49 (±4.69)	53.74 (±4.54)	55.13 (±4.72)	<0.001 ***^1^
**HAO**	[mm]	28.17 (±2.82)	28.14 (±2.93)	28.20 (±2.73)	0.609 ^1^
**CO**	[mm]	16.70 (±3.08)	16.74 (±3.21)	16.67 (±2.97)	0.843 ^1^

^1^ Mann–Whitney U test. (*** *p* < 0.001).

**Table 2 diagnostics-12-02507-t002:** Analysis of age-specific (age in years) differences between offset parameters.

		Total (n = 800)	20–50 y. (n = 384)	>50 y. (n = 416)	*p*-Value
**HO**	[mm]	26.19 (±2.70)	26.07 (±2.61)	26.31 (±2.77)	0.363 ^1^
**GO**	[mm]	61.79 (±5.67)	62.13 (±5.41)	61.47 (±5.89)	0.124 ^1^
**LGHO**	[mm]	54.49 (±4.69)	54.16 (±4.62)	54.79 (±4.73)	0.081 ^1^
**HAO**	[mm]	28.17 (±2.82)	28.17 (±2.80)	28.17 (±2.85)	0.756 ^1^
**CO**	[mm]	16.70 (±3.08)	16.99 (±2.96)	16.44 (±3.17)	0.011 *^1^

^1^ Mann–Whitney U test. (* *p* < 0.05).

**Table 3 diagnostics-12-02507-t003:** Analysis of gender-specific differences between offset parameters.

		Total (n = 800)	Females (n = 351)	Males (n = 449)	*p*-Value
**HO**	[mm]	26.19 (±2.70)	24.17 (±2.01)	27.77 (±2.03)	<0.001 ***^1^
**GO**	[mm]	61.79 (±5.67)	58.14 (±4.52)	64.64 (±4.78)	<0.001 ***^1^
**LGHO**	[mm]	54.49 (±4.69)	50.56 (±3.14)	57.56 (±3.15)	<0.001 ***^1^
**HAO**	[mm]	28.17 (±2.82)	26.34 (±2.14)	29.60 (±2.44)	<0.001 ***^1^
**CO**	[mm]	16.70 (±3.08)	15.92 (±2.80)	17.31 (±3.16)	<0.001 ***^1^

^1^ Mann–Whitney U test. (*** *p* < 0.001).

**Table 4 diagnostics-12-02507-t004:** Analysis of osteoarthritis-specific differences between offset parameters.

		Total (n = 800)	KL 0–2 (n = 777)	KL 3–4 (n = 23)	*p*-Value
**HO**	[mm]	26.19 (±2.70)	26.19 (±2.68)	26.14 (±3.10)	0.941 ^1^
**GO**	[mm]	61.79 (±5.67)	61.85 (±5.63)	59.58 (±6.51)	0.148 ^1^
**LGHO**	[mm]	54.49 (±4.69)	54.51 (±4.66)	53.65 (±5.59)	0.386 ^1^
**HAO**	[mm]	28.17 (±2.82)	28.19 (±2.77)	27.55 (±4.31)	0.571 ^1^
**CO**	[mm]	16.70 (±3.08)	16.75 (±3.04)	15.09 (±3.92)	0.049 *^1^

^1^ Mann–Whitney U test. (* *p* < 0.05).

## Data Availability

All data generated or analyzed during this study are included in this published article. There was no public archievation. Further publications were similarly not made.

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
