# Peer review of "Physiological Offset Parameters of the Adult Shoulder Joint—A MRI Study of 800 Patients"

_diagnostics, 2022, doi:10.3390/diagnostics12102507_

Round 1
Reviewer 1 Report
The paper touches upon an important problem, especially in the practical aspect. I can feel it and know it as a practitioner. Offset assessment, knowledge about it translates into practice - especially in arthroplasty and ORIF, but also in osteotomies.
Despite this, I see a need for improvement:
Abstract; "Humeral (HO) and glenoidal (GO) offset" - should be humeral offset (HO) and glenoidal offset (GO), In the results please specify - "SOP for side and gender in the 40 adult shoulder." in men or women
Introduction: it should be as in the abstract - "Humeral (HO) and glenoidal (GO) offset" - it should be humeral offset (HO) and glenoidal offset (GO), line 69 specify which types of dentures, line 75 emphasize that a shoulder fracture is critical complication
Matology: line 107 - why was the borderline age 20?
Results: it is necessary to standardize the following: the result of p should be to the third decimal place, the rest to the second
Discussion:
more emphasis should be placed on the practical importance of research
Author Response
Reviewer 1:
Thank you very much for your feedback and constructive suggestions for improvement. They were very helpful. After revising the manuscript, we are sure that we have improved the scientific quality of the article with your support.
Point 1:
Abstract; "Humeral (HO) and glenoidal (GO) offset" - should be humeral offset (HO) and glenoidal offset (GO), In the results please specify - "SOP for side and gender in the 40 adult shoulder." in men or women
Response 1:
We thank the reviewer for this advice and corrected this passage. We apologise for the ambiguity and have clarified the phrasing. “For all SOP significantly, higher values were measured in male shoulders. There was a significant (p<0.001) higher mean value for HO, GO and LGHO in right shoulders.” – line 37 ff.
Point 2:
Introduction: it should be as in the abstract - "Humeral (HO) and glenoidal (GO) offset" - it should be humeral offset (HO) and glenoidal offset (GO), line 69 specify which types of dentures, line 75 emphasize that a shoulder fracture is critical complication
Response 2:
We thank the reviewer for this advice and corrected this mistake.
To specify more clearly that both hemi and total endoprostheses are referred to, we have added the following: “Consequently, LGHO reconstruction reduces the risk of early aseptic loosening in hemi- and total shoulder arthroplasty [7,16].” – line 70 f.
We appreciate your advice and have made it clear that a fracture is a serious complication. “The risk of a secondary fracture of the shoulder joint due to an offset change has to be considered critical.” – line 77 f.
Point 3:
Matology: line 107 - why was the borderline age 20?
Response 3:
The initial idea in the study design was to examine patients from six decades. Therefore, the age of 20 years and not 18 years was set as the lower limit for the inclusion criteria.
Point 4:
Results: it is necessary to standardize the following: the result of p should be to the third decimal place, the rest to the second
Response 4:
We thank the reviewer for this advice and standardized all results as recommended.
Point 5:
Discussion:
more emphasis should be placed on the practical importance of research
Response 5:
We have revised the discussion and hope that the clinical significance of the study is now more clearly expressed.
“The clinical consequence of these results would be that, in contrast to the hip joint [26,27], in the shoulder joint for the surgical treatment of severe degenerative diseases or fractures, the healthy contralateral side cannot be used uncritically for planning the offset recon-struction. This hypothesis has to be supported by further studies. However, the data of the present study can be used as a physiological reference to provide an important orientation for the reconstruction of the shoulder offset parameters.” – line 310 ff.
“In conclusion, the newly proposed method for the detection of GO in MRI proves to be re-producible and should therefore be used in clinical practice in the future.” – line 362 f.
“Especially in cases of severe fractures or terminal osteoarthritis physiological reference values are highly important in clinical practice, as they represent an essential orientation with regard to offset reconstruction of the affected joint.” – line 367 ff.
Reviewer 2 Report
There is a major methodological uncertainty in this study. There is no standardized placement of the shoulders (with external standardized references) during the MRI imaging. Therefore the differences of a few millimeters in the measurements might be incidental due to the non-uniform arm position during the tests. Therefore, the differences in the measurements, which are very small, might be influenced by body characteristics. Therefore, the results might be incidental, and the conclusions do not justify publication. Additionally, the tests were done due to shoulder symptoms and could be influenced by muscular spasms during the test therefore the results can’t represent the normal population
Author Response
Reviewer 2:
We thank the reviewer for the justified and constructive criticism. We have modified our manuscript accordingly. In the following, we will comment on it in detail.
Point 1:
There is a major methodological uncertainty in this study. There is no standardized placement of the shoulders (with external standardized references) during the MRI imaging. Therefore the differences of a few millimeters in the measurements might be incidental due to the non-uniform arm position during the tests. Therefore, the differences in the measurements, which are very small, might be influenced by body characteristics. Therefore, the results might be incidental, and the conclusions do not justify publication. Additionally, the tests were done due to shoulder symptoms and could be influenced by muscular spasms during the test therefore the results can’t represent the normal population
Response 1:
We apologise for the lack of information. All patients were placed in a standard shoulder positioning tray on the MRI table.
We have added this point. “In a standardized manner, all patients were placed in a shoulder positioning tray on the MRI table.” – line 109 f.
That physical characteristics could have influenced the measurements is absolutely correct. We cannot rule out muscle spasms for sure either.
The retrospective study design has limitations here. Therefore, the interpretations were toned down in the discussion. These points were recorded under limitations. Furthermore, it was recommended to conduct prospective studies in the future to verify or refute the study results.
The aim of the present study was to conduct a retrospective analysis. Therefore, the results should not be measured by prospective standards.
“The clinical consequence of these results would be that, in contrast to the hip joint [26,27], in the shoulder joint for the surgical treatment of severe degenerative diseases or fractures, the healthy contralateral side cannot be used uncritically for planning the offset reconstruction. This hypothesis has to be supported by further studies. However, the data of the present study can be used as a physiological reference to provide an important orientation for the reconstruction of the shoulder offset parameters.” – line 310 ff.
“No biometric data were collected in the study and the retrospective nature of the study does not allow any conclusions to be drawn about the pain intensity of the patients. Further studies with a prospective design are necessary to verify the study results.” – line 377 ff.
In summary, we agree with you that the study has limitations, also due to the retrospective design. We have now formulated these more clearly.
Nevertheless, we are convinced that the data represent an important reference for future studies and everyday clinical practice. Although further studies are needed for verification.
Reviewer 3 Report
Overall
Summary: In this study, the authors the MRI parameters of adult shoulder joint. However, there are still some points needed to be addressed.
(1) For Abstract
1. Line 39. “No significant differences of HO respectively GO were found in relation to age and osteoarthritis.” What do you mean? Please modify this sentence.
2. Line 41 “An association of HO respectively GO with age and osteoarthritis was not found.” What do you mean? Please make this sentence clearer.
(2) The description of methods is adequate, however, I still have few concerns on this part:
1) Do you consider the body height/body weight into your analysis? Are these parameters of shoulder affected by these two factors?
2) Your results implied that right shoulder has a higher offset (HO,GO, LGHO) than the left shoulder. Do you compare them in the same patient ? If not, I don’t this comparison make sense. Furthermore, the hand dominancy may make influence on the musculature and neurovascular status, although the its influence on the bony geometry is not clear. Do you consider this factor in your analysis?
(3) What’s the clinical implication of the results of this study? How to use the data of this study to modify the clinical practice?
Author Response
Reviewer 3:
We thank the reviewer for the helpful comments and have tried to implement them as best as possible.
Point 1:
(1) For Abstract
- Line 39. “No significant differences of HO respectively GO were found in relation to age and osteoarthritis.” What do you mean? Please modify this sentence.
- Line 41 “An association of HO respectively GO with age and osteoarthritis was not found.” What do you mean? Please make this sentence clearer.
Response 1:
We apologise for the difficulty in understanding and have reworded the sentence. We hope that it is now more understandable. “. There was a significant corelation between age and LGHO, HAO and CO but not between age and HO respectively GO. Shoulders with osteoarthritis and non-osteoarthritis did not differ in the mean value of HO, GO, LGHO and HAO, except for CO (p=0.049).” – line 39 ff.
Point 2:
(2) The description of methods is adequate, however, I still have few concerns on this part:
- Do you consider the body height/body weight into your analysis? Are these parameters of shoulder affected by these two factors?
Response 2:
Thank you for your helpful enquiry. Biometric data was not collected. This aspect was added under Limitations. It is possible that the offset parameters could also be influenced by different biometric data. We find this consideration very inspiring. Prospective studies that clarify this question would be desirable in the future.
“No biometric data were collected in the study and the retrospective nature of the study does not allow any conclusions to be drawn about the pain intensity of the patients. Further studies with a prospective design are necessary to verify the study results.” – line 377 ff.
Point 3:
2) Your results implied that right shoulder has a higher offset (HO,GO, LGHO) than the left shoulder. Do you compare them in the same patient ? If not, I don’t this comparison make sense. Furthermore, the hand dominancy may make influence on the musculature and neurovascular status, although the its influence on the bony geometry is not clear. Do you consider this factor in your analysis?
Response 3:
Thank you for your helpful enquiry.
We did not measure both shoulders of one patient. Therefore, we added following point under limitations: “The handedness of the patients was also not surveyed. This could also have an influence on the offset parameters. Studies dedicated to this question would also be of high scientific value. It should be noted that based on the retrospective study design, no intra-individual bilateral measurements of the shoulder joints were performed. This means that the differences found in the mean values of the SOP between the left and right shoulder could only be determined inter-individually and therefore, have to be viewed critically. In order to be able to make a statement about possible intra-individual side differences for the SOP, fu-ture prospective studies with bilateral joint side analysis should be carried out in order to confirm or revise this observation.” – line 380 ff.
We agree with you that the conclusions drawn have to be weakened due to the limitations. We have tried to implement this and have recommended to verify or refute the results in the future by means of prospective studies.
“The clinical consequence of these results would be that, in contrast to the hip joint [26,27], in the shoulder joint for the surgical treatment of severe degenerative diseases or fractures, the healthy contralateral side cannot be used uncritically for planning the offset reconstruction. This hypothesis has to be supported by further studies. However, the data of the present study can be used as a physiological reference to provide an important orientation for the reconstruction of the shoulder offset parameters.” – line 310 ff.
Studies that include handedness, muscle mass and neurovascular status in the analyses seem useful and highly interesting.
We thank you for your suggestion.
Point 5:
(3) What’s the clinical implication of the results of this study? How to use the data of this study to modify the clinical practice?
Response 5:
We apologise if the clinical benefit of the data has not been made sufficiently clear. We have therefore revised the manuscript and hope that the clinical relevance is now more apparent.
Especially in cases of severe fractures or terminal arthrosis, the results of this study are of high clinical importance in their function as a physiological reference, as they represent an essential orientation with regard to offset reconstruction. From our point of view, the data are an enrichment for the preoperative planning of both joint-preserving and joint-replacing operations.
Furthermore, the newly proposed method for the detection of GO in MRI proves to be re-producible and should therefore be used in clinical practice in the future.
We added these aspects in the end of discussion:
“In conclusion, the newly proposed method for the detection of GO in MRI proves to be re-producible and should therefore be used in clinical practice in the future.” – line 362 f.
“Especially in cases of severe fractures or terminal osteoarthritis physiological reference values are highly important in clinical practice, as they represent an essential orientation with regard to offset reconstruction of the affected joint.” – line 367 ff.
Round 2
Reviewer 2 Report
After the corrections and clarifications can be published